# Character Recognition of Components Mounted on Printed Circuit Board Using Deep Learning

**DOI:** 10.3390/s21092921

**Published:** 2021-04-21

**Authors:** Sumyung Gang, Ndayishimiye Fabrice, Daewon Chung, Joonjae Lee

**Affiliations:** 1Department of Computer Engineering, Keimyung University, Daegu 42601, Korea; smgang.kmu@gmail.com (S.G.); ndayifab2@gmail.com (N.F.); 2Faculty of Basic Sciences, Keimyung University, Daegu 42601, Korea; dwchung@kmu.ac.kr; 3Faculty of Computer Engineering, Keimyung University, Daegu 42601, Korea

**Keywords:** PCB inspection, optical character recognition (OCR), deep learning, coreset

## Abstract

As the size of components mounted on printed circuit boards (PCBs) decreases, defect detection becomes more important. The first step in an inspection involves recognizing and inspecting characters printed on parts attached to the PCB. In addition, since industrial fields that produce PCBs can change very rapidly, the style of the collected data may vary between collection sites and collection periods. Therefore, flexible learning data that can respond to all fields and time periods are needed. In this paper, large amounts of character data on PCB components were obtained and analyzed in depth. In addition, we proposed a method of recognizing characters by constructing a dataset that was robust with various fonts and environmental changes using a large amount of data. Moreover, a coreset capable of evaluating an effective deep learning model and a base set using n-pick sampling capable of responding to a continuously increasing dataset were proposed. Existing original data and the EfficientNet B0 model showed an accuracy of 97.741%. However, the accuracy of our proposed model was increased to 98.274% for the coreset of 8000 images per class. In particular, the accuracy was 98.921% for the base set with only 1900 images per class.

## 1. Introduction

The traditional method of classifying printed characters on components attached to a printed circuit board (PCB) in the past is using machine learning method such as pattern matching. It is still difficult to completely process these characters automatically without human assistance. However, the PCB inspection field is expected to benefit from deep learning because the problem of a low classification accuracy can be solved by converting algorithms from traditional machine learning to deep learning in various fields related to the classification using images.

Deep learning requires enough data for model learning. That data must satisfy both at the same time quantity and diversity requirements. Even when a good deep learning model is trained with a large dataset, there are many differences in performance and accuracy according to the quality of the dataset used for training [1,2,3,4]. When applying deep learning, which can yield high accuracy in general classification problems, it can show good results through collecting, preprocessing, and learning on a large amount of data. However, this is only possible if it is easy to collect data and the collected data is of high quality.

Figure 1 shows an example of a component attached to a PCB, with letters printed on the tops of the components. These components may be visually indistinguishable because each one varies in size, errors that occur when printing and attaching other components, therefore, the inspection can be carried out using images acquired with high-resolution magnifying cameras [3,4,5].

In the past, a pattern matching method of an operator machine was used, which when performing an inspection can save a pattern in advance and compare it with an image. However, this method often results in undetected and misclassified data because many different types of fonts are printed on components, and similar component types often use different fonts. In addition, since each font has a different thickness, during image acquisition, a thin font may be obscured due to a lighting problem.

Applying deep learning for PCB inspection has several problems as follows: Data collection: It is not easy to acquire data during the inspection process because delays in the data collection process can cause decrease the in productivity.Data privacy: Many PCB manufacturers do not allow data collection due to confidentiality issue.Unique problems with the data: Because each PCB production plant uses has different types of PCBs and components used, data collected by one plant can not represent data from all production plants.

In this paper, a representative dataset called a coreset [6] was generated in a manner that could solve unique problems associated with obtaining character data from PCB components. Since actual inspection equipment consists of a limited amount of hardware, deep learning models that can balance measurement time and performance should be utilized. The best EfficientNet model can be selected using the proposed coreset. 

Our work contributes to several things in the process of examining printed characters on parts during PCB testing. The main contributions of this paper are as follows.

We propose a data augmentation method that could generate representative datasets for learning with just data collected from a few factories so that it could be applied to multiple factories for examining printed characters on the top of parts. This method takes color, shape, size, and font shape into account.PCB tester production companies can simultaneously batch-process the learning and distribution and upgrade the deep learning model for multiple factories instead of generating respective models for individual factories.Types of PCB parts continue to grow. Thus, a lot of time is needed to accumulate and learn all collected data. In this work, we used an n-pick grid sampling to include sufficient features while minimizing existing accumulated data, leading to a continuous learning with a quite simple method.

This paper is organized as follows. In Section 2, related studies on deep learning for PCB inspection are discussed and an overview of deep learning models used in this work is given. 

We provide a detailed analysis of the datasets we collected and explain how to reduce and augment the data in Section 3. In Section 4, the most computationally efficient model is selected using the coreset, and a scheme is proposed for constructing a sustainable base set. In Section 5, an experiment that can be considered as a field test is performed; Results of this test can increase the applicability of the proposed method. Finally, we discuss the conclusions of the study and provide future research directions in Section 6.

## 2. Related Work

### 2.1. PCB-Related Optical Character Recognition (OCR) Research and Component Defect Inspection

Regions of the world that are most interested in PCB defect inspection are Southeast Asia, China, and Korea, because industrial production is a major national industry in these regions. By the mid-2010s, deep learning was already actively being used in other research areas, but not in the PCB field [7,8,9,10,11,12]. However, in the past three years deep learning has been actively used in PCB inspection. Such a late adoption of deep learning for PCB inspection was due to various limiting characteristics in this field; for example, database acquisition is difficult and relevant companies are often reluctant to release data. Thus, research papers are often written by companies with PCB production lines [13,14,15,16,17,18,19,20,21].

For OCR work on PCB, the existing industry uses commercial software which uses rotation invariant features or pattern matching for character classification. However, such methods cannot reach a satisfactory level of accuracy In addition, the high license price of the software makes the installed testers less competitive [7,8,9,10,11].

The main purpose of the present study is to use deep learning to classify characters printed on components. Most prior studies have focused on classification of components or application of deep learning for attachment defects of components or defects in PCB circuits [12,13,14] No research has been conducted yet with sufficiently large and diverse databases.

Since each PCB production plant produces different types of PCBs, it is difficult to develop flexible models that can be applied simultaneously to several factories. Consequently, this area of research is still relatively new and has not reached the application phase. Thus, it may take a long time for advancements to be applied in the field.

Although it is difficult to obtain data in the field of PCB defect inspection, DeepPCB [15] might be able to solve some of these problems. DeepPCB is a database of 1500 images, including short and pin-hole defects of PCBs.

Silva et al. [16] have proposed a defect detection method using the DeepPCB database. They showed results were better for shallow models than for deep models. There was no need to use deep models because the DeepPCB images used in their study were binary with simple textures. Accordingly, inherent characteristics of PCB data exist. Thus, it cannot be concluded that a deep learning model with a generally good performance in a study will have the same good performance in the field.

Huang et al. [17] have suggested the use of HRIPCB, a PCB dataset without assembled components, to address the dearth of available data. Indeed, the disclosure of large volumes of data has greatly contributed to the field of study by providing public data that has become a new standard because of an increased accessibility that it provides. While it is true that published datasets can increase the accessibility of research, whether such data are quantitatively satisfactory or representative remains unclear.

PCB production is a very high-speed and precise industry with each factory producing different items. Thus, the representativeness of the dataset must be considered. It is necessary to produce data that can be used as learning data for deep learning models that can then be distributed simultaneously to various production fields. Due to hardware limitations of an inspection equipment used for PCB inspection, online learning techniques are currently not facile. 

Such online techniques aim to collect data from separate factories using inspection machines at the same time and then use the data for training individual deep learning models. Thus, it is necessary to propose a method that ensures that even a small amount of data collected by only a few factories in a short period of time is sufficiently representative.

### 2.2. EfficientNet

EfficientNet [22] was proposed in 2019. It is thought to have sufficient computational speed and accuracy for our application, despite its disadvantage of a slightly slower learning speed compared to ResNet [23].

Generally, model extensions are performed to increase the performance of a convolutional neural network. Also, the deeper the network, the better the performance. If necessary, efficient models can be created by increasing or decreasing the number of nodes. However, making such changes requires a manual input. and a better CNN can be constructed using hyper-parameters after several experiments.

Tan and Le [22] have proposed the best combination of the extension method, using three techniques: (1) increasing the network depth, (2) increasing the channel width, and (3) increasing the resolution of input images.

In the previous study [22], eight models (models B0 to B7) demonstrated superior performance with few parameters when considering optimal combinations within a limited resource range. Figure 2 [24,25] shows the structure of an EfficientNetB0.

## 3. Data Collection and Processing 

### 3.1. Data Collection

Images used in this paper were automatically collected from inspection machines currently used for assembly and inspection lines. The software used during the inspection process was capable of separating and labeling characters from strings printed on faces of components. 

However, the software could not separate each character because the distance between characters was noticeably short. Moreover, when labeling separated character images, there was an error of specifying a different label. Figure 3 shows a problem that cannot be properly divided because of the short distance between characters [3,4,5].

All data were automatically labeled using a pattern matching software that was previously used for PCB inspection during the collection process. If the software were 100% accurate, our research would be unnecessary.

Even if the labeling process of the software is simply a substitute for some extremely hard work by humans, there are many errors because they are not accurate. Therefore, we performed re-labeling by manually checking. This meant that both learning data and testing data were manually validated by our researchers. 

### 3.2. Class Definition

Collected data is defined by a total of 52 classes in this research. There were originally a total of 62 classes corresponding to different characters on faces of PCB components, consisting of 10 digits (0–9), 26 uppercase letters (A–Z), and 26 lowercase letters (a–z). However, a total of 20 labels of the same shape but different sizes were combined and reduced to 10 labels (i.e., C and c, K and k, O and o, P and p, S and s, U and u, V and v, W and w, X and x, and Z and z). Therefore, in this study, there were a total of 52 classes, comprising 10 numbers and 42 letters.

### 3.3. Analysis of Data Imbalance Issues

Data were collected for learning and testing purposes in consultation with the company that manufactured the inspection equipment. A large set of segmented text images consisting of a total of 5,368,069 massive images was acquired from some lines of over five assembly plants, because PCB production and processing occurred amazingly quickly. However, there were large differences in the collected number and style of fonts in each class because not all components used the same font.

When deep learning models learn with a dataset, the model will certainly show good results in the factory where the dataset is collected. However, it is unreasonable to expect good results for factories that use components with different fonts on printed surface. It not only takes a long time to collect data and train models if different models need to be applied to different PCB inspection sites, but this also becomes practically impossible because each site must then employ deep learning experts who understand unique characteristics of various datasets.

To address these problems, a company that manufactures the inspection equipment must apply a deep learning model to the equipment with a long-term data collection plan. Maintenance also needs to be considered. Figure 4 shows the distribution of data for a total of 52 classes. In this distribution, the amount of data acquired per class was highly imbalanced. No data had been collected for classes that were not frequently used for components. For example, the total number of numeric classes from 0 to 9 was approximately 360,000, which accounted for almost 70% of the overall total.

The most used class, 0, accounted for about 13% of all data. If most classes had so much data, we could use the same method of down-sampling for all classes. However, classes such as X, Z, and e deserved attention. 

Classes with a small amount of data can simply generate as much additional data as possible through up-sampling. However, this process can cause overfitting because some classes only contain roughly the same image as the cloned image.

Figure 5 shows examples of image types for several classes. For 0 and A, six examples are shown because there are many different image styles. For i, O, e, X, Y, and Z, all image types in each class are shown. All 1551 images in the i class have the same style. The same is true for all 815 images in the O class.

### 3.4. Enhancement and Augmentation of Character Data

In this section, data are augmented and enhanced in five aspects to overcome limitations of having a small amount of data that is highly redundant. These methods available in this case are up-sampling, accomplished by transforming existing data and generating unpredictable new data from existing data.

Diversity in illumination, rotation, size, noise, and contamination (occlusion) damage could be addressed through variations in collected data. In addition, new data can be created in response to unknown data styles (i.e., new color and new font types) by considering both font shape and colors of the foreground and background. The method of augmentation of existing data cannot be applied to a few classes (classes with only one kind of image or no data). Existing data augmentation methods could create new data by interpolating existing data, or simply add noise, rotate, and invert. However, since there was no letter in the shape of a flipped ‘e’ and since flipped ‘d’ becomes ‘b’, it is difficult to use a method of flip the letter to train a model. 

That is, the flip method to augment dataset could make the meaning of a letter become completely different from that of the original letter. In addition, since completely new data created were not included in collected data from factories, it was difficult to overcome the bias with an augmentation method that could slightly transform pixels of existing data. For that reason, a large portion of this study is focused on generating new data.

#### 3.4.1. Illumination Variance

Text data obtained from PCB components are expected to mostly consist of similar colors. However, a given color could differ slightly depending on the color of the component or the environment in which it was acquired. Some types of characters are obtained only from certain components, which might vary depending on the factory from which data were acquired. This could result in a foreseeable character recognition problem due to, rotation, illumination, noise error, and new font shapes.

Figure 6a shows data of various colors. Figure 6b illustrates the process of reversing colors in each image. This result was obtained by subtracting the maximum color of each RGB channel and then setting it to an absolute value. The data set was constructed such that the deep learning model could learn as many types of distributions as possible. A few images exhibiting variance in illumination were mixed into the coreset because it was relatively unlikely that data with colors shown in Figure 6b would exist without such intervention.

Figure 7 briefly illustrates the distribution of data in several classes. Although the i class had only one image type, the distribution was somewhat widely spread due to differences in the number of pixels in different images caused by; the large amount of data in that class. On the other hand, for the e class, there were 512 data points. The distribution was very narrow.

#### 3.4.2. Rotation Variance

Some of the collected data might have been rotated slightly by attached components. The font might have also been rotated. The angle of the rotation was never more than +5 degrees or less than −5 degrees as rotation by more than a certain angle would result in an attachment error. 

Consequently, considering the environment in which the learned model was to be applied, some collected data were randomly rotated at an angle between +5 and −5 degrees, as shown in Figure 8.

#### 3.4.3. Size Variance

Because the sizes of components vary, the size of the collected data was also different for each image. Some components were exceedingly small. Thus, characters written on them could be small enough to cause problems for learning models. In most images used in this study, outer edges of characters were removed. A square size was changed to 32 × 32 pixels for deep learning training. Sizes of original images varied widely, with a few being more than 100 pixels in width or length, while some being less than 32 pixels in length. If the image was enlarged to 64 × 64, which was four times the size of 32 × 32, this was not a meaningful process because pixels were simply duplicated to the enlarged part during the interpolation process Thus, the images on which the model was trained were 32 × 32 pixels in size, as in the CIFAR-100 dataset.

#### 3.4.4. Noise and Contamination (Occlusion) Variance

In this study, data were augmented using random Gaussian noise, a common component of the augmentation process. To address damaged data, data were augmented for the purpose of responding to it as shown in Figure 9a. In the process of acquiring data, various types of damage could occur. For example, data might be partially cut off or damage could be caused by printing and heat treatment of fonts on the surface of the component. We replaced instances of degradation with partial noise, as shown in Figure 9b. In addition, most of damaged data were collected separately from the process described in the previous section (Section 3.1), and the data imbalance problem was solved by adding collected data to the coreset.

#### 3.4.5. Font Shapes Within Class Variance

The data augmentation process mentioned in the previous section did not completely resolve the data imbalance problem because it required up-sampling with a finite image. As shown in Figure 4, some fonts were not collected at all. In the opinion of inspection equipment developers and industry experts, such classes cannot be excluded from model learning because they might be used for printing component faces in the future. 

Data augmentation methods such as generative adversarial networks (GAN) [26] have been widely used in recent years. They can be used to create new data by fully learning the information from existing collected data. 

For some cases, none of their data was collected from whole factories (i.e., for lowercase f, m in Figure 4). Thus, even with methods such as GAN, we could not generate labeled data [27,28].

To solve this problem, a total of 35 free font images were used to generate new font images as shown in Figure 10. This method was since letters printed on parts attached to the PCB were from the font installed on a computer [29].

The method of doing so involved the following steps: (a)Fonts installed on the computer (35 types of fonts in this study) was used to generate 52 classes. It meant that 52 characters’ images were generated for 35 types of fonts, resulting in a total number of 1820 images. The foreground of the image was made black and the background was made white.(b)This was repeated for a total of 52 times (from 0 to Z) and 35 font styles were randomly selected.(c)The RGB channel of the image was converted to hue, saturation, value (HSV) channel [29].(d)When R, G, and B values of a pixel were 0, it was a foreground (i.e., a character). H, S, and V values of a pixel were randomly set to satisfy the following ranges: 0 ≤ H < 30, 100 ≤ S < 256, and 100 ≤ S < 256.(e)Conversely, if R, G, and B values of a pixel were 255, it was a background. H, S, and V values of the pixel were randomly set to satisfy the following ranges: 0 ≤ H < 30, 100 ≤ S < 256, and 0 ≤ S < 150.(f)Processes (4) and (5) were repeated by checking all pixels in the image. In addition, ranges for H, S, and V in processes (4) and (5) were just examples. They were empirically set as color frequently found in PCBs.(g)The HSV channel image was converted to RGB and saved.(h)Random noise was performed for half of processes from (2) to (7) and mixed into the image.(i)The operation from processes (2) to (8) was repeated as many times as set by the user. In this work, this method was utilized to create 8000 images per class.

We created HSV range conditions for the background and foreground of processes (3) and (4) in several empirical ways, allowing us to randomly select data when generating them to produce as many different images as possible. To ensure that created images were similar to images extracted from real PCBs, unnecessary borders were cut out of final images.

RGB channels of the image can be converted to HSV channels by Equation (1) [29], where pixels R, G, and B have values between 0 and 255.
(1)$$V=\max\left(R,G,B\right)\;S=\left\{\begin{array}{c} \frac{V-\min\left(R,G,B\right)}{V}\;\;if\;V\neq 0\\ \;\;\;\;\;\;\;\;\;\;\;\;\;\;0\;\;\;\;\;\;\;\;\;\;\;\;\;\;\;\;if\;V\neq 0 \end{array}\right.\;\;H\;\left\{\begin{array}{c} \frac{60\left(G-B\right)}{V-\min\left(R,G,B\right)}\;if\;V=R\\ 120+\frac{60\left(B-R\right)}{V-\min\left(R,G,B\right)}\\ 240+\frac{60\left(R-G\right)}{V-\min\left(R,G,B\right)} \end{array}\right.\;if\;H<0,\;H=H+360\;$$

### 3.5. Data Sampling and Grid Algorithm

#### 3.5.1. Overfitting and Underfitting

Overfitting means over-learning the data. It means when the deep learning model is fit too tightly for the learning data. It occurs when data have an excessive bias. In general, learning data are a subset of real data. Thus, errors decrease for learning data but increase for real data. Learning data are a subset of real data because it is impossible to collect all real data. On the other hand, underfitting is a state in which the deep learning model does not fully reflect characteristics of the learning data. It can occur when learning time is too short or when the expressiveness of deep learning models is lower than the level required to represent the data.

Data collected in this study were too dense for certain shapes and forms before augmentation. Although some data we collected had an exceptionally large number, they were highly biased and imbalanced. Another class had no data at all or the number of data was exceedingly small. 

Learning data is a subset of data that can occur in an environment where deep learning is applied. It is exceedingly difficult to learn data from a small number of factories and apply it to most factories. Because characteristics of each class are completely different, it is difficult to learn to predict data that the model has never learned before no matter how good the deep learning model is. If we learn a deep learning model by using our data as it is, it can be overfitted for certain factories. However, the model is not fitted for other factories.

Figure 11 [30] illustrates underfitting, overfitting, and optimal fitting. In Figure 11, the yellow line, red points, and blue line represent the original curve, data sampled from the curve, and curve fit to red points, respectively. From the perspective of the data, in general, large amounts of data can prevent overfitting. However, even if there are a lot of data like our data, data with a narrow distribution and extremely poor quality could not prevent overfitting. Therefore, a model fitted to the poor quality sample may not fit well to the population. In other words, problems can arise because the diversity of collected samples is less than the data diversity of the actual environment in which our learning model is applied.

Although some data used in this study were exceptionally large, the distribution of data tended to be dense for some classes, while there were no data at all or only a few types for data in other classes. Consequently, overfitting and underfitting could occur for each class. To solve the parameter fitness problem of the deep learning model, various types of data were created in the process of augmentation and reinforcement as discussed in the previous section. To balance the amount of data in each class, it is necessary to perform appropriate up-sampling or down-sampling. However, it was difficult to apply the same sampling method to all classes because data collected from PCB had imbalances within and between classes, as explained previously [30,31].

The SMOTE [32] algorithm is a commonly used oversampling technique that can generate synthetic data. It can solve the problem of overfitting by sampling several classes and interpolating a few samples to synthesize a new instance. However, this method does not fit for some data that we need to address. For example, for the worst class in our set (i.e., ‘X’), up-sampling is virtually meaningless. This was because only a small change in pixel values was made from the original data that was repeatedly collected, although additional data were created through data augmentation. Therefore, we need to augment our data for new and unfamiliar data predictions and reduce the amount of data by selecting images that are as different in style as possible [6,30,31,32,33,34,35].

#### 3.5.2. Grid Sampling

The easiest way to know that each image has a different style is to analyze pixels in the image. We have proposed a grid sampling method of pixel analysis in previous studies [3,4,5]. This method can be called restrictive random sampling as it limits the sampling location. The left side of Figure 12 shows the distribution of the original and augmented data for class ‘0′ and, the right side of the figure shows results after scaling of the data using the n-pick (n = 3) method of the grid algorithm. After applying the grid algorithm, the distribution was no longer concentrated in certain spots. Instead, it was much more even.

In the graph, each point represents an image. The *x*-axis is determined by obtaining m in Equation (2) while the *y*-axis is determined by obtaining σ in Equation (2). At this time, V is the brightness channel of the image obtained from Equation (1), w is the width of the image, and h is the height of the image:(2)$$m=\frac{\sum_{i}^{w\times h}Vi}{w\times h}\;,\;\;\;\;\;\;\sigma =\sqrt{\frac{\sum_{i}^{w\times h}{\left(Vi-m\right)}^{2}}{w\times h}}$$

The data to be handled is an image in which the background and foreground are clearly separated. The average of pixels may vary depending on the shape of the font. For example, even for multiple images having same ‘0’ shape of font, the foreground will occupy a larger proportion if a thick font is used. 

Also, if the foreground or background color is different, the deviation will be different. The distribution shown in Figure 12 sufficiently takes the style of the image into account.

The grid is created by dividing the *x*-axis and *y*-axis by 5. Since images distributed in one grid have similar lightness distributions, the font type and color tone will be similar. Therefore, if there are little data in one grid, all data will be included in the down-sampled data. In contrast, if there are too much data in the grid, overfitting can be avoided by taking only a few data points.

Algorithm 1 is the pseudocode of the grid algorithm proposed in [3] and [4] and the n-pick algorithm additionally proposed in this study. Data can be extracted by specifying the number per class (e.g., 8000 images) or by specifying the number of n in one grid according to the n-pick condition.

**Algorithm 1:** Grid and n-pick sampling algorithm***imgNum = number of images******n = Number of n for n-Pick sampling******m = Number of grid sampling that extracts a total of m sheets from the original dataset******imageList = Data list of Total images (image path)******randomPickupImg = random sampling algorithm****graph_w = 255, graph_h = 150, cut_num = 5**grid_w = int(graph_w/cut_num), grid_h = int(graph_h/cut_num)****/* Check the number of data in each grid */******for** i to imgNum:* ***for** x to grid_w:*  ***for** y to grid_h:*   ***if** mean[i] >= x*cut_num **and** mean[i] < (x + 1)*cut_num:*    ***if** std[i] >= y*cut_num **and** std[i] < (y + 1)*cut_num:*     *countNumberOfImgInGrid[x,y] + = + 1****/* Calculate the percentage of data occupied by each grid */****imgRatioInGrid[x,y] = (countNumberOfImgInGrid[x,y]/imgNum)****/* N-pick sampling of one grid */******if** nPick grid sampling:* ***for** x to grid_w:*  ***for** y to grid_h:*   ***if** countNumberOfImgInGrid[x,y] > 0:*    *finalSampledDataList = randomPickupImg (imageList in grid [x. y]) **until** n****/* Grid sampling for a total of m sheets */******else:*** ***for** x to grid_w:*  ***for** y to grid_h:*   ***if** countNumberOfImgInGrid[x,y] > 0:*    ***if** (m *  imgRatioInGrid[x,y]) > 1:*       *z = (int) (m *  imgRatioInGrid[x,y])*    ***else if** (m *  imgRatioInGrid[x,y]) < 1:*       *z = 1*    *finalSampledDataList = randomPickupImg (imageList in grid [x. y]) **until** z* ***if** Number of finalSampledDataList > m:*  *randomPickupImg (finalSampledDataList) **until** m*
***/* Save image in sampling folder */***

*Save(finalSampledDataList into sampling folder)*


At this time, data can be sampled in two ways: typical grid sampling considering how many will be extracted from the original images when extracting data (i.e., variable ‘m’ in Algorithm 1) or n-pick sampling. For example, if there were 1000 original images, users could decide whether to extract 100 or 10 images. In other words, it is a method of reducing data while keeping data density of each grid before and after sampling.

Assume that the total number of original images is 10,000 and that we want to reduce the amount of data to 1000 images by using the grid algorithm. In this case, if the number of images in a one grid before sampling is 1000 images and 10% from the total amount of original images, the percentage of images after grid sampling is also 10% from the total amount of sampled images. That is, 100 images will be in that grid after sampling.

The n-pick sampling does not consider the number of final sampled images at all. It only considers n numbers in one grid. This method allows the sampled image to be evenly distributed throughout the graph. If a deep learning model is trained with the data extracted through n-pick sampling, the model can learn different types of images as much as possible.

## 4. Proposed Method

### 4.1. Process of the Proposed Method

In this work, data augmentation was performed with sufficient consideration of the properties of data in databases with severe imbalances. Assuming that a deep learning model should be learned and applied to multiple environments using augmented data, the most effective model considering CPU and GPU should be selected, and the accuracy of the model should be verified using a method in which a sustainable set of bases can be constructed. Figure 13 shows the proposed process which has two steps: (1) The best model is selected by verifying B0 to B7 models of EfficientNet learned with grid-sampled data using 8000 images per class (called the coreset in this study) on the CPU and the GPU; (2) The proposed n-pick(n = 3) base set in this study is verified with the selected model and contrasted with the comparison target. Details are shown as follows.

Each EfficientNet model from B0 to B7 was trained on 8000 images per class defined by grid sampling. The accuracy and speed were measured during each test. While minimizing defects is particularly important in the real world, replacing the inspection equipment already in use on site with a computer equipped with a high-end graphics card is more costly than the loss caused by the presence of defects. 

In addition, it is impossible to replace all hardware at once. Therefore, the best models that show an appropriate balance between performance and computational speed should be applied on site. The verification should be carried out on both the CPU and the GPU. The number of inspections and the production count per second should be estimated when the deep learning model is applied in different hardware environments.

The base set can be verified using an effective model selected after the previous step. Several targets are selected for comparison. Whether the base set could provide a sustainable dataset proposed in this study was assessed. Targets for comparison were (1) the result of 8000 images per class, and (2) 12,000 images per class. 

The n-pick (n = 3) base set was constructed by extracting three images per grid, as shown in Figure 12 using the grid sampling method described in Section 3.5.2. Since various combinations of databases have been performed in our previous work [3,4,5], we only analyze the grid and n-pick algorithms in depth in this work.

We have identified several points in the previous study [3,4], and the learning data in the previous study has been subdivided by case, and the test data is as shown in Table 1.

Models learned using sample data from few factories cannot be represented by models for all factories.As a result of experimenting with combinations of all cases, the accuracy is higher when mixing newly generated data using fonts than existing data augmentation.When using grid sampling algorithms, the higher the number of sampled data, the better the learning accuracy. However, considering the learning speed, about 8000 pcs are effective.Previous studies did not mention the n-pick algorithm.

### 4.2. Configuration of the Coreset

Data were organized as shown in Table 1. Data collected from two plants were classified into datasets 1 and 2. Generated font data was named dataset 3. The test dataset consisted of a mix of data collected from plants 1 and 2 at different time points and much more data collected from other plants. In general, a deep learning model evaluates the performance with 20% of the data after separating the collected data into learning data (80% of total data) and testing data (20% of total data). 

In our experiment, the most important goal was to produce a model applicable to other production lines even if it is formed on a limited amount of data from a small number of production lines.

Thus, the collection period and location of the test dataset in our study were different from those of the learning dataset. We believe that this process is well worthwhile as a field test before the model is applied to the real world. Although a true field test would involve testing by an actual inspector, our experimental design was more effective than performance evaluation in a limited laboratory environment.

The basic combined dataset is based on data collected from two production plants. In other words, data collected from two factories were mixed for each class, and 8000 images for each class were extracted. When there was no or insufficient data for a given class, the number of images was increased by forcibly up-sampling the data with noise. For classes without data at all, all final images originated from dataset 3. Therefore, after combining necessary data from dataset 3 into the data from datasets 1 and 2, 8000 sheets per class were extracted by grid sampling. In addition, 12,000 sheets per class were extracted and organized in the same manner. In the n-pick (n = 3) dataset, n images were extracted per grid, which could be configured to minimize redundancy.

Table 2 shows the number of images used in each process. Since there were 8000 sheets per class, the total was divided at a ratio of 8:2 to form learning and validation sets. In the n-pick (n = 3) dataset, the amount of data per class was, approximately 1900 images on average. The number of data images per class differed slightly because it was based on the distribution of the grid.

## 5. Implementation and Experimental Results

Experimental learning was conducted in a multi-GPU environment. We compared results produced with a CPU and a single GPU and evaluated the performance of each of teight EfficientNet models. The experimental environment had an Intel^®^ Core™ i7-6700K @ 4.00 GHz 4.01 GHz CPU, 64 GB RAM, Multi GPU graphics card (two GeForce GTX TITAN X), and Tensorflow 1.13.1 backend Keras 2.2.4-tf.

To assess the general situation before the experiment, results of each model were checked without sampling the actual data. The use of unsampled data meant that all collected data could be put into training without considering data imbalance between classes.

We trained three models (ResNet 56 layers, EfficientNet B0, and EfficientNet B7) and confirmed the prediction results. Input data had 32 × 32 pixels in size. Model training occurred for a total of 50 epochs. Results are shown in Table 3.

In this experiment, the model was trained using dataset1 (see Table 2). All collected data were inserted without considering the data imbalance between classes. This experiment revealed class-specific and intra-class data imbalances. It was possible to see results of learning with unsampled and highly biased data. Thus, only dataset 1 at plant 1 were selected on a trial basis without dataset 2 at plant 2. This experiment did not use dataset3 because its data were augmented data.

As shown in Table 3, results of the recently proposed EfficientNet model are particularly good, although there is still room for improvement. The ResNet results made it difficult to say that this learned model was worth using as it was. Its results were poor compared to those of EfficientNet. Results of the B0 and B7 EfficientNet models were not much different from each other. This meant that there was no significant difference in the performance between B0 and B7 models when learning was done with an unmodified set of collected data. These findings enabled us to evaluate the performance improvement that was possible by using the coreset defined in our experiment.

### 5.1. Selection of Models with High Accuracy and Good Computation Speed

Prior to training the model for field applications, we allowed the 56-layer ResNet model to learn for comparison purposes and checked the results, which are shown in Table 4. Compared to Table 3, results shown in Table 4 were quite different, despite the difference between whether sampling was performed, and several data augmented methods were used. In particular, the ResNet56 model showed an accuracy of 97.763% (Table 4) in a model trained with augmented data and sampled data, although its performance was poorer than that of the EffcientNet model.

Table 3 shows results of training with the ResNet 56 model using more quantitative but unsampled data (original dataset1). Table 4 shows results of learning using data extracted from 8000 images per class by grid sampling after combining original datasets (No.1 and No. 2) and augmented data generated. The test dataset for checking the accuracy was the same.

The augmented dataset was more efficient than the original dataset as illustrated by the fact that analyzing the data based on the distribution with the V channel was meaningful.

In this experiment, the top five most accurate results were combined because some characters were confusing from a visual perspective as shown in Figure 14. For such cases, there might be errors even if a person was manually performing the classification. When applied to an industrial site in the future, results can be calculated by integrating these ambiguous classes into a single class after computation.

In other words, this problem cannot be completely controlled by humans. In the end, the use of standardized fonts in the PCB industry is required. However, it is difficult to elicit consultations between various production companies in each country.

Fortunately, the PCB assembly has a design document. Letters printed on the part can also be known because it can be known which parts are plugged into which location in advance.

Thus, after deep learning operations, labels such as 0 and O, B and 8 are reported as simple programming methods (if-elseif, etc.) and combined to confirm them. For example, if the third character printed on that part is ‘B’, the model prediction result after deep learning will be ‘8’ as the first choice with ‘B’ as the second choice. This can be processed as the correct answer.

To put it simply, it can be considered that when training a pre-deep learning model, it is possible to extract the result by combining two classes into the same class (i.e., ‘B’ and ‘8’ into the same class). However, a problem can occur because shapes of some characters such as ‘B’ and ‘8’ are similar for only a few font types. For other font types, such characters are completely different. Therefore, classes were separated to prevent confusion in learning considering various situations.

Table 5 shows a comparison between the CPU and the GPU speed. PCB assembly and inspection equipment might have GPUs (embedded graphics cards) that do not support deep learning frameworks or have no GPU at all.

By comparing the computational time needed for two types of equipment, it is possible to assess the applicability of deep learning to PCB defect inspection in the real world. The accuracy of a deep learning model performed in a CPU environment may be higher than that of a machine learning method such as pattern matching. The speed may also increase. Therefore, it is necessary to judge whether the number of PCB production per day decreases due to the calculation speed instead of increasing the accuracy. If the inspection speed is slightly lowered but manual process by humans is decreased by a high accuracy, then the number of PCBs produced will be approximately the same.

All eight EfficientNet models were trained and their results were compared. Since the model used a small image (32 × 32 pixels) as the input data, a model that was too deep such as B7 could be considered inefficient. In addition, the feature map extracted by the deep neural network from a small image might be redundant. However, EfficientNet extracted feature maps very effectively, with B7 showing the best result.

Table 6 shows results with each EfficientNet model. EfficientNet model B3 with the lowest accuracy outperformed ResNet which had an accuracy of 97.763%. Notably, although EfficientNet models, which already showed good performance, demonstrated significant increases in accuracy (an increase of 0.533%pt from 97.741% to 98.274% for B0 and 1.174%pt from 97.887% to 99.065% for B7). Since differences between models B0 to B7 were insignificant, it could be said that the influence of the coreset especially important. Even a difference in accuracy of only 1%pt represents 3675 images in a total dataset of 367,510 images. In the real field where many PCBs are produced and inspected in a short time, a 1%pt difference in performance has a large influence on productivity.

Table 7 shows the CPU and the GPU speeds for each EfficientNet model. As the 56-layer ResNet model had a speed of 0.00606 s, we expected to observe improvements in both accuracy and speed when using an EfficientNet model below B5. As shown in Table 7, models from B0 to B5 had speeds faster than 0.00606 sec, the speed of ResNet 56 layer in CPU. In Table 6, Model B1 showed the highest accuracy of 99.003%. 

Therefore, the application capability may vary depending on the situation at the site or the judgment of a professional worker. According to results shown in Table 6 and Table 7, B1 model had the best result based on top one and top five accuracies.

### 5.2. A Continuously Available Base Set for Future Learning 

This experiment was performed using datasets that were constantly changing due to factors such as an upgrade of the inspection equipment. Our proposed base set can prevent the distribution of data from being crowded on either side when performing new learning. Since the number of components was constantly increasing, training and model application ware not one-time tasks. Instead, tasks had to be repeated continuously.

Two things should be considered in continuous learning. First, performing prior experimental procedures only on newly collected data will create the problem of missing information from existing data. Second, experimenting with a combination of both existing and newly collected data can result in continuous data growth and increased the processing time. Thus, using the n-pick dataset proposed in this study is a proven way of mixing data directly into continuously collected datasets.

Using more data generally produces better results. Thus, we conducted tests using 12,000 images per class and compared results to those produced using the n-pick dataset proposed in this study. Results are shown in Table 8.

When 12,000 images were used per class, the learning data were overwhelmingly large. Thus, the experiment (on 499,200 total images) produced the highest accuracy. Most importantly, although the n-pick dataset was only about one-sixth as large, results are also highly accurate. With 8000 images per class, the B1 model resulted in an accuracy of 99.003%. The accuracy of the B1 n-pick (n = 3) experiment was 99.275%, showing an improvement of 0.272%pt. This is amazingly effective in terms of learning time and accuracy because the use of only about a quarter of the data can produce better results.

N-pick sampling creates a slight imbalance in the number of images between classes. However, as shown in Figure 15, the use of n-pick sampling can flatten data distribution because the method extracts data of as many different shapes as possible within the class. Thus, it is possible to include new information while retaining existing information with the proposed n-pick data sampling process.

### 5.3. Analysis of Results and Failure Factors

In this section, precision, recall, and f1-score were analyzed in detail for experiments using 8000 images per class in order to analyze the B7 model, which showed the best result among EfficientNet models. Based on results of the analysis, a class that may be problematic is selected and future measures are considered.

As can be seen in Table 9, each performance indicator showed a good overall performance. However, the macro average did not show a good performance. The micro average was the average of all classes. The macro average was the average of the average per class. A low score of a macro average means that some classes have a problem below dropping the average value. 

Figure 16 shows actual error cases and identifies problems expected in the previous section. Some mistakes are easy for a human to make. There is a similarity between several “A” and “R” characters. Several truncated or damaged characters warrant attention in the future. New data augmentation methods should be considered.

## 6. Conclusions

In this work, we addressed dataset generation learning and application for classification of characters printed on top of real PCB components. We collected large datasets and we made suggestions for data augmentation methods to be referenced in future studies with more detailed analysis of the data. Moreover, we proposed a combination of various datasets to select a coreset that could be used in the field and a base set that could be used continuously in the future.

In our method, the B7 model using 8000 images per class had an accuracy of 99.065. When combining similar classes into the same class, the top five accuracy was 99.965%, showing an increase of 1.178%pt in accuracy compared to the case when the model learned on data collected from a single factory. Considering the amount of test data, this model was very efficient in terms of productivity because it predicted about 4400 more images correctly. Furthermore, we presented a measure to ensure that existing information is not forgotten when an ever-increasing amount of learning data were added, as shown by the accuracy result of 99.275% for the B1 model with only about 1900 base sets per class.

In the future, it is necessary to consider methods that can deal with damaged datasets and similarities among different fonts. To correct for truncated characters, we intend to expand upon this research by focusing on a deep learning model such as GAN that generates data.

## Figures and Tables

**Figure 1 sensors-21-02921-f001:**
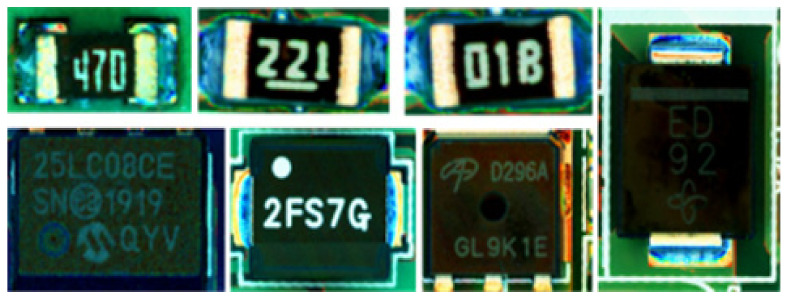
Examples of components attached to PCBs.

**Figure 2 sensors-21-02921-f002:**
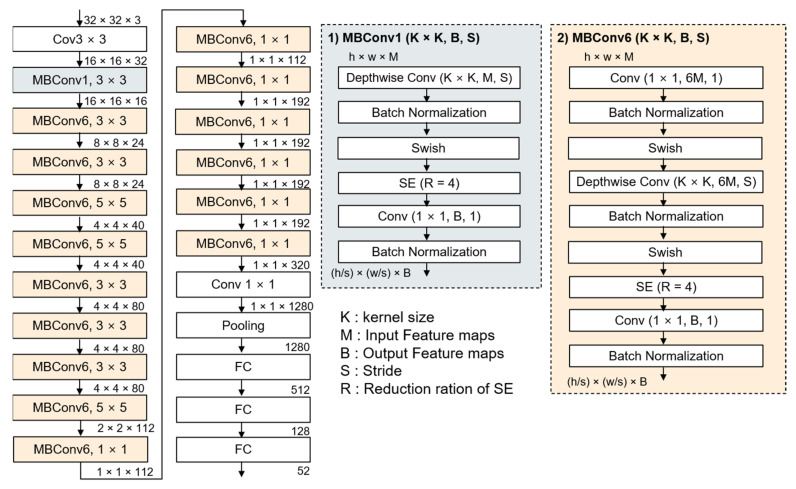
The structure of an EfficientNetB0 model with the internal structure of MBConv1 and MBConv6. Compared to MBConv1, MBConv6 has three layers at the top. The number of feature maps as the output is 6.

**Figure 3 sensors-21-02921-f003:**
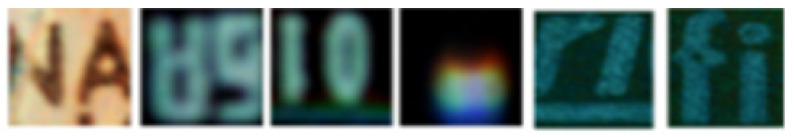
Examples of images with errors.

**Figure 4 sensors-21-02921-f004:**
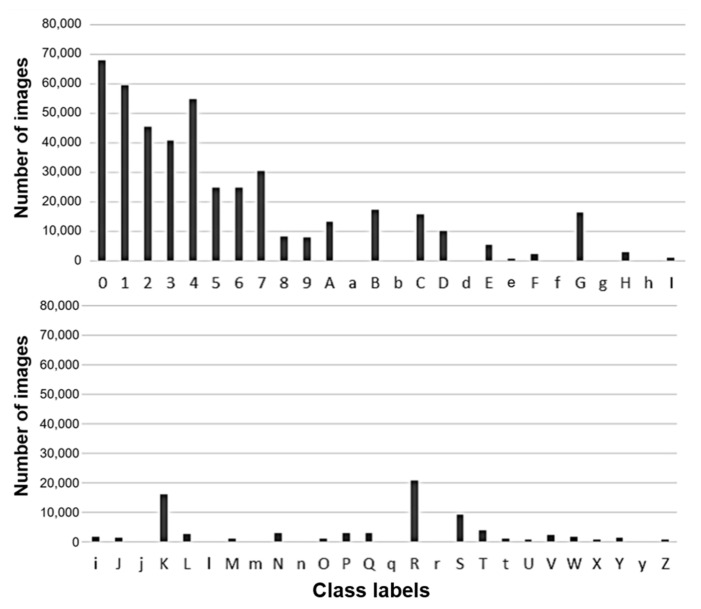
Data distribution by collected character class.

**Figure 5 sensors-21-02921-f005:**
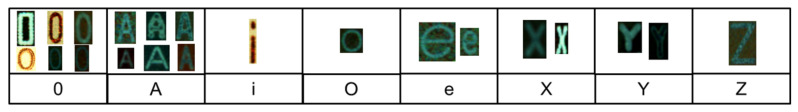
Examples of image types in each class.

**Figure 6 sensors-21-02921-f006:**
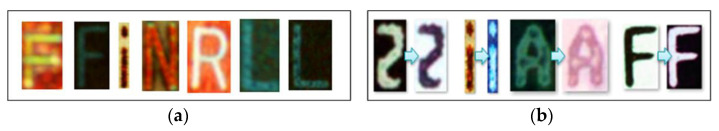
(**a**) Data in various colors, (**b**) Result of color reversal.

**Figure 7 sensors-21-02921-f007:**
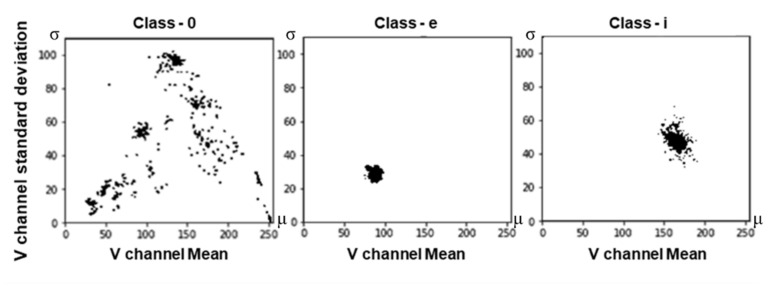
V channel distribution: showing the mean number of pixels and standard deviation for classes 0, e, and i.

**Figure 8 sensors-21-02921-f008:**
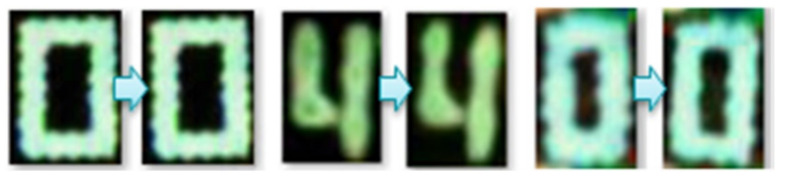
Random rotation of data at an angle within ±5 degrees.

**Figure 9 sensors-21-02921-f009:**
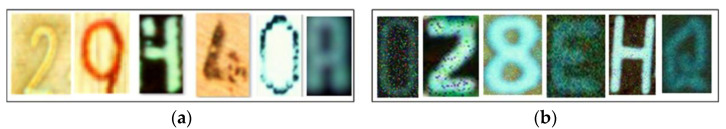
(**a**) Example of corrupted data (**b**) Example of noisy data.

**Figure 10 sensors-21-02921-f010:**
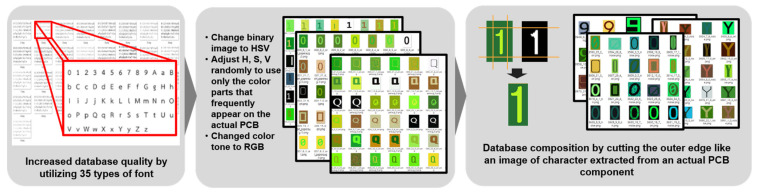
Creation of font data for colors appearing on PCB components in the real world.

**Figure 11 sensors-21-02921-f011:**
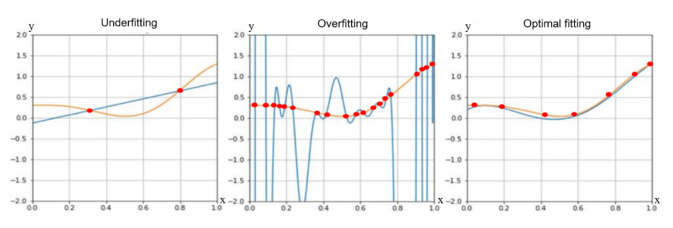
Underfitting, overfitting, and optimal fitting.

**Figure 12 sensors-21-02921-f012:**
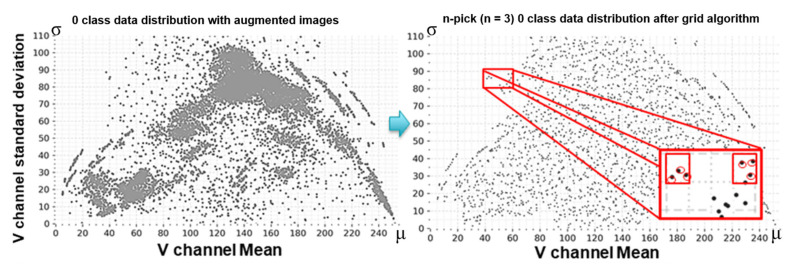
Data reduction using the grid algorithm. The left part of the figure is a mixture of both original data 1, 2, and augmented data in class 0. The right part of the figure shows the result after applying the n-pick sampling algorithm.

**Figure 13 sensors-21-02921-f013:**
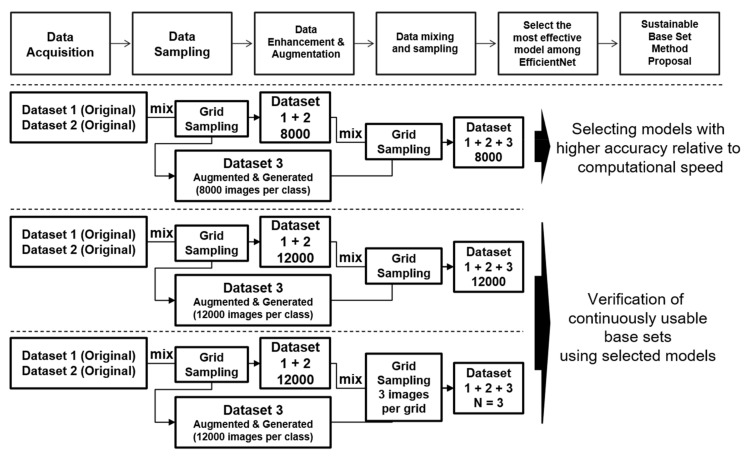
The processes of the proposed method.

**Figure 14 sensors-21-02921-f014:**
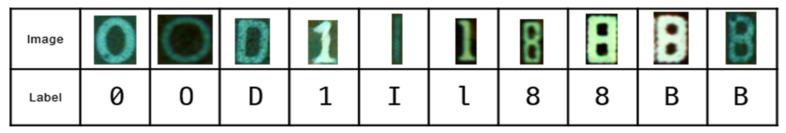
Examples of ambiguity between classes.

**Figure 15 sensors-21-02921-f015:**
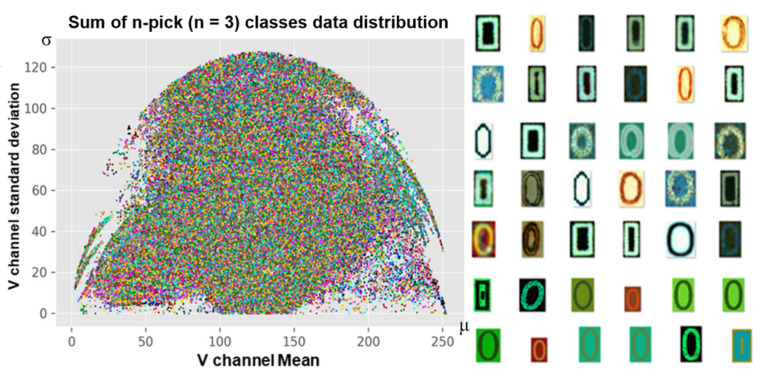
Visualization of distribution for all classes of the n-Pick (n = 3) dataset and an example of sampled data of class 0.

**Figure 16 sensors-21-02921-f016:**
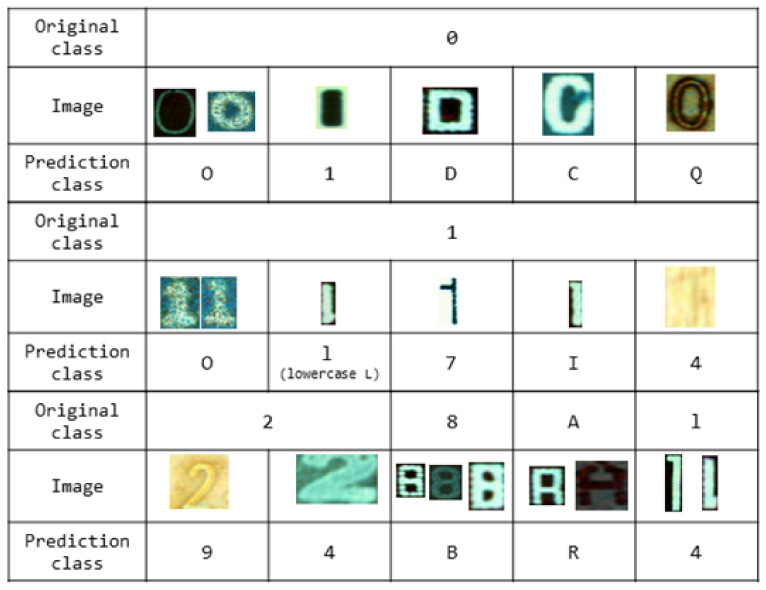
EfficientNet B7 error cases with 8000 images per class.

**Table 1 sensors-21-02921-t001:** Database description.

No	Database	Details
1	Dataset 1	Collected data from the 1st factory
2	Dataset 2	Collected data from the 2nd factory
3	Dataset 3	- Augmented images from datasets 1, 2 - Generated font data
4	Test dataset	- Collected data from the 1st factory, 2nd factory, and other factories (over five factories) - Amount of data from the 1st factory, 2nd factory: 10% of the total amount - Amount of other factories data: 90% of the total amount - Collect at a different period

**Table 2 sensors-21-02921-t002:** Database description.

No	Images Per Class	Number of Learning Data	Number of Validation Data	Number of Test Data
1	8000	332,800	83,200	367,510
2	12,000	499,200	124,800	
3	3 images per grid	79,395	19,848	

**Table 3 sensors-21-02921-t003:** Experimental accuracy (%) on dataset 1.

Experiments	ResNet 56 Layers	EfficientNet B0	EfficientNet B7
Original dataset1	79.917	97.741	97.887

**Table 4 sensors-21-02921-t004:** ResNet 56 experiment: top 1 and top 5 accuracy (%) results with 8000 images per class.

Experiments	ResNet 56 Layers Top1	ResNet 56 Layers Top5
8000 images per class	97.763	99.9

**Table 5 sensors-21-02921-t005:** CPU vs. GPU computational speed (Sec.) per image for each model in ResNet.

Category	CPU	GPU
ResNet 56	0.00606	0.00028

**Table 6 sensors-21-02921-t006:** EfficientNet Experiment showing top 1 and top 5 accuracies with 8000 images per class (%).

No.	Experiments	EfficientNet Top1	EfficientNet Top5
1	B0	98.274	99.881
2	B1	99.003	99.965
3	B2	98.869	99.955
4	B3	98.135	99.900
5	B4	98.643	99.955
6	B5	98.395	99.942
7	B6	98.965	99.940
8	B7	**99.065**	**99.965**

**Table 7 sensors-21-02921-t007:** Average calculation rate for EfficientNet experiments (s).

Category	EfficientNet Models							
	B0	B1	B2	B3	B4	B5	B6	B7
CPU	0.00120	0.00170	0.00183	0.00235	0.00326	0.00479	0.00644	0.00883
GPU	0.00027	0.00036	0.00037	0.00043	0.00055	0.00071	0.00083	0.00113

**Table 8 sensors-21-02921-t008:** Average calculation rate for EfficientNet experiments (s).

No.	Experiments	EfficientNet B0 Top1	EfficientNet B1 Top1
1	12,000 images per class	99.829	99.851
2	n-pick(n = 3) dataset	98.921	99.275

**Table 9 sensors-21-02921-t009:** Results of EfficientNet B7 with 8000 images per class.

Category	Precision	Recall	F1-score	Support
micro avg	0.9904	0.9904	0.9904	367,510
macro avg	0.7067	0.7339	0.7158	367,510
weighted avg	0.9922	0.9904	0.9912	367,510

## Data Availability

The data presented in this study are available on request from the corresponding author. The data are not publicly available due to the request of the company that provided the data.

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
