# Peer review of "Character Recognition of Components Mounted on Printed Circuit Board Using Deep Learning"

_sensors, 2021, doi:10.3390/s21092921_

Round 1

Reviewer 1 Report

  1. In Figure 2, the image resolution quality needs to be improved. Besides, if the upper right of "MBConv6 (k3x3)" in Figure 2 is the magnification of the third step [MBConv1, 3x3], it is necessary to provide easy illustrations, including "MBConv5, (k5x5)".
  1. We appreciate the use of data augmentation, even though it is currently popular with GAN (Generative Adversarial Network). In sub 3.4.5, How to use 35 different fonts to create an image for each of 52 classes should be given an algorithm/pseudo like a grid algorithm.
  1. If class ambiguity becomes a problem in this deep learning, it will be better to be followed up at least to determine the accuracy of this class. What is the follow-up from Figure 14?
  1. In Table 3, the trained dataset is only Dataset 1 using ResNet-56, EfficientNet B0, and EfficientNet B7. What will the results be for the other two datasets?
  1. What are the differences between the results in Tables 3 and 4 for the model of ResNet 56 layers?
  1. What are the variables of the horizontal and vertical axes?
  1. Please briefly describe the n-pick method and V channel distribution in the paper.
  1. Does it need the parentheses for the term wxh in equation (2)?

Author Response

Thank you very much for your sincere review. Also, thank you for the time and effort you took to review the article. Based on your comments, the title and direction of our article have been changed and the overall paper has been improved. All concerned lines of the article in this comment are located at revision version of “track changes of MS Word” Detailed responses to your comments are shown below :

MAJOR

  1. In Figure 2, the image resolution quality needs to be improved. Besides, if the upper right of "MBConv6 (k3x3)" in Figure 2 is the magnification of the third step [MBConv1, 3x3], it is necessary to provide easy illustrations, including "MBConv5, (k5x5)".

=> We have improved the image resolution quality as you requested (line172). Also, some errors were found and corrected. MBConv6 was not an extension of MBConv1. We have added some explanation to the image. where 1 and 6 are related to the number of feature maps as output.

  1. We appreciate the use of data augmentation, even though it is currently popular with GAN (Generative Adversarial Network). In sub 3.4.5, How to use 35 different fonts to create an image for each of 52 classes should be given an algorithm/pseudo like a grid algorithm.

=> The method of generating data using fonts is a simple method that does not require algorithms. We have added more explanation to make it clearer to readers. We also explained the method in lines 341-387 so that the process could be a little clearer per your suggestion. Thank you very much for your insightful comments and suggestions.

  1. If class ambiguity becomes a problem in this deep learning, it will be better to be followed up at least to determine the accuracy of this class. What is the follow-up from Figure 14?

=> We thank the reviewer for this your suggestion, and we agree with the reviewer. Therfore, we have mentioned these issues in lines 626-642.

  1. In Table 3, the trained dataset is only Dataset 1 using ResNet-56, EfficientNet B0, and EfficientNet B7. What will the results be for the other two datasets?

=> Results of the original dataset 2 were similar because the distribution of the data was biased. We replaced these comparisons with only one experiment. This content may be ambiguous. Thus, we revised the paper as shown in lines 588-591.

  1. What are the differences between the results in Tables 3 and 4 for the model of ResNet 56 layers?

=> Table 3 shows the result of learning using the ResNet 56 model with quantitatively larger but unsampled data (original dataset1). Table 4 shows the result of learning using data extracted from 8000 images per class by grid sampling after combining original datasets No. 1, No. 2, and augmentation data. Test data set for accuracy verification was the same. This was written in the paper as shown in lines 604-613.

  1. What are the variables of the horizontal and vertical axes?

=> We thank the reviewer for pointing this out and we agree with the reviewer. Therefore, we have provided variables for the horizontal and vertical axes of the picture and corrected all errors in the picture.

  1. Please briefly describe the n-pick method and V channel distribution in the paper.

=> We thank the reviewer for pointing this out and we agree with the reviewer. Therefore, we have provided a brief description for the n-pick method and the v-channel distribution in the revised version of the manuscript (Section 3.5.2, added lines 489-502).

  1. Does it need the parentheses for the term wxh in equation (2)?

=> Yes, thank you for your suggestion. we have inserted parentheses for the term wxh.

Reviewer 2 Report

The paper discusses an important practical applications, however there are some important issues related to the paper:
1. An important part of the paper is the proposed resampling method. The justification of the method as well as its description is quite unconvincing. 

E.g. authors state that "Overfitting is a problem if there is too much
data in one area. However, underfitting can occur if the amount of data that is available is insufficient [25]" (l.305-307), while it is normally adopted that overfitting happens if we have not enough data (or have too little variability in the data) with respect to the number of model parameters - which is rather the opposite. Note that similar statement (l. 297-298) is very unclear.

Since the re-sampling methods form a core of the paper scientific contribution, they must be discussed in-depth and rich referenc to existing bibliography should be provided. 

The differnce between 'normal' grid sampling and n-pick sampling is not well described. E.g. it is not explained why using n-pick sampling we obtain less samples. The Algorithm 1 doesn't explain the difference. According to the pseudo-code in normal sampling mode we (possibly?) pick one element per grid, while in 'n-pick' we pick n-elements. However it is hard to tell for sure because it is not explained how randomPickupImg(int) operates. Additionally coundImgInGrid is unitialized, samplenum is not well explained, there are also some typing errors such as 'nPick==ture'

2. It is not explained how the training and test data are labelled. Whether they are labelled manually or automatically by an OCR system. If the second is the case it should be given how this system compares to the system developed and what about wrong labels provided by the OCR system?

3. Main paper contributions are not appropriately described. Vague remarks from 63-80 should be replaced by clearly stated improvements over state-of-the-art and previous self-cited works [3],[4].

4. The problem solved is not clearly stated in the introduction. The introduction should provide information on problem backgroud and problem statement, now it starts from some unclear discussion regarding big-data.

5. Experiments on some public dataset should be performed and results compared to competetive methods.

6. Evaluation on sample set coming exclusively from different production sites (than training data) would be more convincing than evaluation on mixed data.

7. Some sententces are very unclear:

l. 92-93.  'Commercial software was mainly used for OCR, the main algorithms of which include rotating robust feature technicians or pattern matching.'
l. 151-153. ' However, as there were problems with character separation and labeling errors, the data shown in Figure 3 was deleted or cut into single-character images [3,4].'
l. 208-211 'The method of augmentation of existing data cannot be applied to a few classes (classes with only one kind of image or no data). For that reason, a large portion of this study is focused on generating new data.' (why cannot be applied?)
l. 253-254 'pixel crushing could occur during interpolation.' (explain what is pixel crushing)
l. 464-465 'However, increasing accuracy naturally increases computational speed, so user judgment is important'
l. 487-488 'but the model with the best results when judging by top 1 and top 5 is B1 according to the results in Tables 6 and 7 combined.' - not clear how they are combined...

7. Language should be seriously revised, currently the paper is hard to read and quite hard to understand.

Author Response

Thank you very much for your sincere review. We also thank you for the time and effort you spent reviewing the article. The title and direction of our article have been changed and the overall paper has been improved. . All concerned lines of the article in this comment are located at revision version of “track changes of MS Word” Detailed responses to your comments are shown below :

The paper discusses an important practical applications, however there are some important issues related to the paper:

  1. An important part of the paper is the proposed resampling method. The justification of the method as well as its description is quite unconvincing.

E.g. authors state that "Overfitting is a problem if there is too much

data in one area. However, underfitting can occur if the amount of data that is available is insufficient [25]" (l.305-307), while it is normally adopted that overfitting happens if we have not enough data (or have too little variability in the data) with respect to the number of model parameters - which is rather the opposite. Note that similar statement (l. 297-298) is very unclear.

Since the re-sampling methods form a core of the paper scientific contribution, they must be discussed in-depth and rich referenc to existing bibliography should be provided.

The differnce between 'normal' grid sampling and n-pick sampling is not well described. E.g. it is not explained why using n-pick sampling we obtain less samples. The Algorithm 1 doesn't explain the difference. According to the pseudo-code in normal sampling mode we (possibly?) pick one element per grid, while in 'n-pick' we pick n-elements. However it is hard to tell for sure because it is not explained how randomPickupImg(int) operates. Additionally coundImgInGrid is unitialized, samplenum is not well explained, there are also some typing errors such as 'nPick==ture'

=> We thank the reviewer for pointing this out and we agree with the reviewer that there is an error in our description. Therefore, we have revised definitions of overfitting and underfitting. We additionally wrote that we were concerned about the problem of overfitting and underfitting in view of data. Thank you for your valuable comments. We have revised the manuscript as shown in lines 393-428. We also confirmed that there was a big error in Algorithm 1. Thank you very much for your careful comment. The algorithm is now described in lines 488-502.

  1. It is not explained how the training and test data are labelled. Whether they are labelled manually or automatically by an OCR system. If the second is the case it should be given how this system compares to the system developed and what about wrong labels provided by the OCR system?

=> We thank the reviewer for pointing this out and we agree with the reviewer that the data labeling and collection process are ambiguous. Therefore, we have added that part to lines 188-200.

  1. Main paper contributions are not appropriately described. Vague remarks from 63-80 should be replaced by clearly stated improvements over state-of-the-art and previous self-cited works [3],[4].

=> We thank the reviewer for pointing this out and we agree with the reviewer. Therefore, we have summarized and modified what we actually did as shown in lines 79-95. Our research is strong in terms of industrial application. Some expectations are reflected.

  1. The problem solved is not clearly stated in the introduction. The introduction should provide information on problem backgroud and problem statement, now it starts from some unclear discussion regarding big-data.

=> We thank the reviewer for pointing this out and we agree with the reviewer. Therefore, we have revised this part in lines 28-36

  1. Experiments on some public dataset should be performed and results compared to competetive methods.

=> We thank the reviewer for pointing this out and we agree with the reviewer. Our study only considered data collected from PCBs in a limited way. Unfortunately, public datasets do not have this similarity. In addition, public font data collected from PCBs are unavailable. Thus, we analyzed our results through various comparisons in the research process. Our limitation is that it is impossible to compare our results with public data in our experimental design process.

  1. Evaluation on sample set coming exclusively from different production sites (than training data) would be more convincing than evaluation on mixed data.

=> We thank the reviewer for pointing this out and we agree with the reviewer that better results can be obtained for proprietary factory data when learning the model. However, applying different models for each plant will have time and cost issues. In addition, not all factories have deep learning experts. Moreover, companies that manufacture and sell inspectors cannot produce models for each company due to labor shortages. The biggest consideration of industry application is associated with cost. Therefore, real industrial sites require models that can be shared across multiple factories. This content has been fully explained across several lines of the paper.

  1. Some sententces are very unclear:

l.92-93. 'Commercial software was mainly used for OCR, the main algorithms of which include rotating robust feature technicians or pattern matching.'

=> (Please check L.117-121)

 l. 151-153. ' However, as there were problems with character separation and labeling errors, the data shown in Figure 3 was deleted or cut into single-character images [3,4].'

=> (Please check L.188-196)

l. 208-211 'The method of augmentation of existing data cannot be applied to a few classes (classes with only one kind of image or no data). For that reason, a large portion of this study is focused on generating new data.' (why cannot be applied?) => (Please check L. 260-267.)

l. 253-254 'pixel crushing could occur during interpolation.' (explain what is pixel crushing)

= >(Please check L. 313-314)

l. 464-465 'However, increasing accuracy naturally increases computational speed, so user judgment is important'

=>(Please check L. 650-656)

l. 487-488 'but the model with the best results when judging by top 1 and top 5 is B1 according to the results in Tables 6 and 7 combined.' - not clear how they are combined...

=>(Please check L. 679-687)

=> We thank the reviewer for pointing this out and we agree with the reviewer. Therefore, we have revised them accordingly.

  1. Language should be seriously revised, currently the paper is hard to read and quite hard to understand.

=> We thank the reviewer for pointing this out and we agree with the reviewer. Therefore, we have asked a Professional English Editing Service (Harrisco) to improve the language of this paper.

Round 2

Reviewer 2 Report

General remark: The line numbers provided in the answers do not match the actual text, which makes the verification of corrections and improvements quite challenging...

ad. 1 
discussion regarding overfitting, underfitting and data balancing must be improved:

"there is a long time for learning" - l. 353 - early stopping is one way to mitigate overfitting, but it does not mean that long training is the cause of overfitting....
"(the data) they were highly biased" - what does it exactly mean? how do we learn that they were biased?
"it can be overfitted for certain factories but underfitted for other factories" (370-371) - the data for two domains (factories) may have different structure and the model can overfit 1-st domain, but it does not mean that the second domain is 'underfitted'
"From the perspective of the data, overfitting is a problem if there are too much data in one area (i.e., highly biased)" (l. 375) - I cannot agree with this - having more data prevents overfitting. Presumably the authors want to counteract bad data balance in some areas of space (where data do not correspond to the distribution in the population), but it has little to do with underfitting and overfitting. Underfitting and overfitting is inherently a problem of too simple or too complex model with regard to the underlying data 'structure' and not really the data 'size' (although more data may help to overcome problems with overfitting because it is harder to fit the noise if there are more samples....).

ad. 2 
it is still not explained, how the data were labelled (using OCR or manually?) (at least not in lines 188-200), so the question still applies

ad. 6. 
This does not answer the question. The could be rephrased as: if it was verified experimentally, that the system trained on Factory 1 can be transferred to Factory 2 without re-training?

Other remarks:

  1. Since the system strongly relies on data preparation it would be good to see a comparison in one table (and for the same set of parameters and network architectures), how the system compares for original, augmented and grid-filtered training data....Now this kind of comparison requires comparing multiple tables.
  2. The color-wheel from Fig. 10 seems to have Wikipedia origin - this should be referenced.
  3. l.352-358 - it should be better to use 'training' instead of 'learning' data
  4. Although English proof-reading was performed the language is still hard to understand for some parts of the paper...
    'the learning data are extremely lower than the diversity of data" l. 376-377
    'If the total number of images before sampling was 1000 and the density of a grid was 10% (100 images), the grid density was also about 10% (about 10 images) after being sampled with 100 images. It might change slightly considering other grids. However, the overall distribution follows the existing distribution.' l. 435-438
  5. Review the paper for typos e.g. >>l. 386<<, >>'if coundImgInGrid[[x,y] > 0:' in Algorithm 1 <<
  6.  l. 319-340 - it is rather not a good idea to mix bullet points with enumeration

Author Response

Reviewer 2 Comments (for 2nd round):

Thank you very much for your review.

Below please find our point-by-point response.

General remark: The line numbers provided in the answers do not match the actual text, which makes the verification of corrections and improvements quite challenging...

= > We thank the reviewer for pointing this out and we apologize for this. This was because "track changes" caused the problem. Thus, we mentioned both line numbers with and without track changes in the answer,

1) L.T. : Lines applied with ‘track changes’ status in MS Word

2) L. : Lines without “track changes” status

  1. 1

discussion regarding overfitting, underfitting and data balancing must be improved:

"there is a long time for learning" - l. 353 - early stopping is one way to mitigate overfitting, but it does not mean that long training is the cause of overfitting....

= >The part has been deleted. (L. 353, L.T.548 )

"(the data) they were highly biased" - what does it exactly mean? how do we learn that they were biased?

=> This means that data "between classes" and "within classes" are not balanced. we have explained why our data are highly biased in Sections 3.2, 3.3, and Figure 7). We evaluated the bias by considering it in terms of font shape or pixel value. We considered the bias in terms of font shape or pixel value of constructed images.

Please Refer to Fig. 4 for imbalance between classes. Some data are too much and some data are not collected at all. Also, please refer to Fig. 5 and Fig. 7. For imbalance within the class, in Fig. 7, Class 0 has a wide distribution, but class e or i has fewer data styles (Fig. 5) and a narrow distribution ( Fig 7). We judged these points were due to highly biased data.

The part has been revised to "they were highly biased and imbalanced." (L. 362-363 / L.T.415-416 )

"it can be overfitted for certain factories but underfitted for other factories" (370-371) - the data for two domains (factories) may have different structure and the model can overfit 1-st domain, but it does not mean that the second domain is 'underfitted'

= > The part has been revised to " If we learn a deep learning model by using our data as it is, it can be overfitted for certain factories. However, the model is not fitted for other factories " (L. 370-371 / L.T. 423-424)

"From the perspective of the data, overfitting is a problem if there are too much data in one area (i.e., highly biased)" (l. 375) - I cannot agree with this - having more data prevents overfitting. Presumably the authors want to counteract bad data balance in some areas of space (where data do not correspond to the distribution in the population), but it has little to do with underfitting and overfitting. Underfitting and overfitting is inherently a problem of too simple or too complex model with regard to the underlying data 'structure' and not really the data 'size' (although more data may help to overcome problems with overfitting because it is harder to fit the noise if there are more samples....).

=> Yes, we agree with you in the meaning of the phrase. “Too much data prevents overfitting.”

A lot of that data can prevent overfitting is because data have to satisfy both quantitative and qualitative compositions. However, our data have a special distribution. Thus, the model trained with our data is overfitting with specific data distribution.

The phrase has been changed so that there is no misunderstanding. (L. 374-377 / L.T. 432-435)

  1. 2

it is still not explained, how the data were labelled (using OCR or manually?) (at least not in lines 188-200), so the question still applies

=> We mentioned the method of data collection and labeling in section 3.1. The labeling was done using OCR program as the 1st and manually as the 2nd (L. 169-175 / L.T.191-203 )

  1. 6.

This does not answer the question. The could be rephrased as: if it was verified experimentally, that the system trained on Factory 1 can be transferred to Factory 2 without re-training?

 => Yes, it is possible. Please look at Table 1. Our learning data are 1, 2, and 3. The test data is number 4. Number 4 (test data) consisted of 1st and 2nd factory data that accounted for 10 % of the total test data. On the other hand, other factories data accounted for 90% of the total data. In our research, when only factory 1 data were used to train the model, results for the test data are shown in Table 3. it showed a quite bad result. However, results of 1+2+3 shown in Tables 4 and 6 are better. In other words, the model learned with factory 1 (with augmentation and applied grid algorithm) data is applicable to factory 2 without retraining.

Other remarks:

Since the system strongly relies on data preparation it would be good to see a comparison in one table (and for the same set of parameters and network architectures), how the system compares for original, augmented and grid-filtered training data....Now this kind of comparison requires comparing multiple tables.

=> Thank you for your comments. We totally agree with your opinion. Since our study is an extension of our previous work [3, 4], we have performed comparisons for many cases in that study. This means that all data configurations and combinations have been previously validated. Since various combinations of databases have been performed in our previous work[3,4], we analyze the grid and n-pick algorithms in depth in this work. We have added the information for that part (L.475-477 / L.T.564-565).

The color-wheel from Fig. 10 seems to have Wikipedia origin - this should be referenced.

=> Thank you for your comments, we have added a reference [28].

l.352-358 - it should be better to use 'training' instead of 'learning' data

=> Thank you for your comments. All training data have been changed to learning data.

Although English proof-reading was performed the language is still hard to understand for some parts of the paper...

=> Thank you so much for pointing this out and we agree with your. We have asked a professional English Editing company (HARRISCO) to improve our English.

'the learning data are extremely lower than the diversity of data" l. 376-377

=> Thank you for your careful review. That part has been revised (L.377-380 / L.T.435-439).

'If the total number of images before sampling was 1000 and the density of a grid was 10% (100 images), the grid density was also about 10% (about 10 images) after being sampled with 100 images. It might change slightly considering other grids. However, the overall distribution follows the existing distribution.' l. 435-438

=> Thank you for your careful review. That part has been revised (L.437-442 / L.T.437-441).

Review the paper for typos e.g. >>l. 386<<, >>'if coundImgInGrid[[x,y] > 0:' in Algorithm 1 <<

=> Thank you for your careful review. That part has been revised (L.386 / L.T.449 ) / (Algorithm 1).

  1. 319-340 - it is rather not a good idea to mix bullet points with enumeration

=> We thank the reviewer for pointing this out and we agree with the reviewer. That part has been replaced by a number (L.318-339 / L.T.351-371).

Again, thank you for giving us the opportunity to strengthen our manuscript with your valuable comments and queries. We have worked hard to incorporate your feedback and hope that these revisions persuade you to accept our submission.

Round 3

Reviewer 2 Report

The latest answers and paper improvements are generally satisfactory. However, I will provide some short discussion that
may help to improve the paper.

1. Your answer: = > The part has been revised to " If we learn a deep learning model by using our data as it is, it can be
overfitted for certain factories. However, the model is not fitted for other factories " (L. 370-371 / L.T. 423-424)

The cited phrase is not exactly what we see in the paper (at least in my version).

2. paper - l. 377-378 "On the other hand, underfitting may occur when the sample diversity is much less than the population's data diversity."

I am not quite sure - this still looks like overfitting (learning noise which is the result of non-representative sampling).
Overfitting happens when the system does not generalize well to unseen data and works well for the training data - and I think this is still the case here...

3. Your answer

=> Yes, it is possible. Please look at Table 1. Our learning data are 1, 2, and 3. The test data is number 4. Number 4 (test data) consisted of 1st and 2nd factory data that accounted for 10 % of the total test data. On the other hand, other factories data accounted for 90% of the total data. In our research, when only factory 1 data were used to train the model, results for the test data are shown in Table 3. it showed a quite bad result. However, results of 1+2+3 shown in Tables 4 and 6 are better. In other words, the model learned with factory 1 (with augmentation and applied grid algorithm) data is applicable to factory 2 without retraining.

OK. This is important information confirming that the system is able to generalize well and thank you for including this data in the table 1. Please note that the table 1 may be hard to read since the rows are not well aligned.

4. Your answer:

=> Thank you for your comments. We totally agree with your opinion. Since our study is an extension of our previous work [3, 4], we have performed comparisons for many cases in that study. This means that all data configurations and combinations have been previously validated. Since various combinations of databases have been performed in our previous work[3,4], we analyze the grid and n-pick algorithms in depth in this work. We have added the information for that part (L.475-477 /L.T.564-565).

I still suggest summarizing your results in one place for better readability. 

Author Response

Reviewer 2 Comments (for 3rd round):

Thank you very much for your review.

Below please find our point-by-point response, and I marked the changed phrase in red in my revised paper. 

The latest answers and paper improvements are generally satisfactory.

However, I will provide some short discussion that may help to improve the paper.

  1. Your answer: = > The part has been revised to " If we learn a deep learning model by using our data as it is, it can be overfitted for certain factories. However, the model is not fitted for other factories " (L. 370-371 / L.T. 423-424)

The cited phrase is not exactly what we see in the paper (at least in my version).

= > Thank you for your careful review. There was a mistake that I had uploaded a different version of the file. That part has been revised (L.370-372).

  1. In paper - l. 377-378 "On the other hand, underfitting may occur when the sample diversity is much less than the population's data diversity."

I am not quite sure - this still looks like overfitting (learning noise which is the result of non-representative sampling).

Overfitting happens when the system does not generalize well to unseen data and works well for the training data - and I think this is still the case here...

= >Thank you for your suggestion. That part was revised at L. 378-381, as “Therefore, a model fitted to the poor quality sample may not fit well to the population. In other words, problems can arise because the diversity of collected samples is less than the data diversity of the actual environment in which our learning model is applied.”

  1. Your answer

=> Yes, it is possible. Please look at Table 1. Our learning data are 1, 2, and 3. The test data is number 4. Number 4 (test data) consisted of 1st and 2nd factory data that accounted for 10 % of the total test data. On the other hand, other factories data accounted for 90% of the total data. In our research, when only factory 1 data were used to train the model, results for the test data are shown in Table 3. it showed a quite bad result. However, results of 1+2+3 shown in Tables 4 and 6 are better. In other words, the model learned with factory 1 (with augmentation and applied grid algorithm) data is applicable to factory 2 without retraining.

  1. This is important information confirming that the system is able to generalize well and thank you for including this data in the table 1. Please note that the table 1 may be hard to read since the rows are not well aligned.

= > Thank you very much. I fully understood that the table has an alignment problem. The height of the table has been increased to make it easier to understand. (table1)

  1. Your answer:

=> Thank you for your comments. We totally agree with your opinion. Since our study is an extension of our previous work [3, 4], we have performed comparisons for many cases in that study.

This means that all data configurations and combinations have been previously validated. Since various combinations of databases have been performed in our previous work[3,4], we analyze the grid and n-pick algorithms in depth in this work.

We have added the information for that part (L.475-477 / L.T.564-565).

I still suggest summarizing your results in one place for better readability.

= > Thank you for your comments, we have added summarizing the result of previous research [3,4] at L. 482-491.

Again, thank you for giving us the opportunity to strengthen our manuscript with your valuable comments and queries. We have worked hard to incorporate your feedback and hope that these revisions persuade you to accept our submission.

This manuscript is a resubmission of an earlier submission. The following is a list of the peer review reports and author responses from that submission.

Round 1

Reviewer 1 Report

                  Dear Editor and Authors,
     I have read the paper entitled "Coreset Construction and Basic Dataset Configuration for Character Recognition and Inspection of PCB Parts using Deep Learning", which deals with a very interesting topic relevant for readers interested in CNNs and PCB manufacturing for the electronic industry.
     Content-wise, although I am not a specialist in CNNs (I accepted to review the paper as most probably have not read fully the title and somehow didn't register the "Deep Learning" as I saw first the "PCB") I believe it could be a good paper and potentially a very useful one (and I must commend the Authors for the significant amount of time and effort they must have put it for the many simulations and trials which must have been carried out to obtain the validation and performance qualification numbers for their method). However, as I started reading and going through the paper it became clear that it has -in my personal opinion- some important flaws related to its content. Therefore, my decision is to request for MAJOR corrections, as follows:

MAJOR
-  The English needs to be drastically improved in many places throughout the entire paper, e.g.:
- at p.1 line 4 bring on the previous line the comma with which line 4 starts (probably you need to delete a blank space between "components" and the comma), ;                                                                  

- at p.1 lines 7 & 9 replace "coreset" (or "coresets") with "core set" (or "core sets"), and likewise at p.1 line 22;
- at p.1 line 16 replace "quantitatively high, but qualitatively low" with "quantitatively present in large numbers, but most of them are of low quality, i.e. poor readibility,"

- at p.1 line 17 replace "data collected" with "data are collected"
- at p.1 line 18 replace "shapes and tones" with "sizes, shapes and tones for the letters/text to be recognized"
- at p.1 line 18 I presume that the Authors meant "addresses", not "find". Also, at the end of the same line/sentence, the Authors should indicate the reference for their "previous study".
 - at p.1 line 19 replace "low quality and large amount of" with "large amounts of low quality"
    And these are the corrections ONLY for the first page of the manuscript!! Many other English phrasing corrections must be made throughout the entire paper, which -for brevity- will no longer be detailed here. I strongly suggest the Authors to forward the paper (AFTER finalizing all its corrections) to a professional English editing & proofreading service before resubmission.

 - Related to Fig.1: again the English is not optimal, and I think the Authors should rephrase its caption in a simpler manner just stating that very few fonts are used/printed on the devices present on a given manufacturer's PCBs.
     Related to Fig.2: I don't understand why the need to identify each and every combination of letters and denominate it as a "class". I thought the main target was to first correctly identify/recognize each SEPARATE letter (as hinted in the caption of Fig.1 with the singular example for the zeros!), since once one achieved this, making up their combinations as written on the device will be trivial. Moreover, this may simplify the whole operation, since all the characters (i.e. letters & numbers, probably plus a few other symbols) are well known and their number is ALWAYS FIXED, instead of having a theoretically infinite number of possible character combinations (practically can be limited, but still larger than the number of just the characters in the alphabet + the numbers, AND this number of combinations may also be potentially fluctuating/modifying in time as other devices may be used instead for new products). Indeed, later on, in Section 2, the Authors seem to focus only on recognizing the letters & numbers, but then this should be made very clear from the very beggining in the paper, when the "class" must also be clearly and unambigosly defined and described.

 - At p. 2 lines 31-32: Why is it difficult to collect databases from PCBs? And how exactly did the Authors managed to overcome this difficulty and "collect enough data"? The Authors should provide much more detailed info about each of these issues.
    Later, at line 34, what is exactly is "the imbalance of the data"? Does it refer to frequency of different types of character combinations?? If yes, then it should be clearly indicated as such, as well as explained why is this a "problem". On one hand, obviously certain devices are used in much larger numbers than others (e.g. the number of resistors with "10 k" inscription on them may siginficanly outnumber the number of microprocessors with their own different inscription), so this "problem" clearly must be expected and thus dealt with (How do the Authors intend to handle it?). On the other hand, however, the "problem" wouldn't even appear in the first place if the focus were on the correct identification/recognition of individual characters, with which then their combinations can be easily determined (hence there would be no problem as well with using the model in any factory) which does seem to tbe focus of the Authors, but then if they intend to recognize the separate characters, why would their many separate combinations in classes matter at all?? I don't understand why the statistics of all the "class names" (in itself a bad name, in my opinion, each is just a particular set/combination of characters) is also important and/or relevant for their analysis.
     Moreover, what is (are) the key difference(s) that differentiate the core set from all the "classes"? Why is it "balanced"? How is it arrived at/deduced? It would be much simpler and more efficient to ensure first and foremost the effective correct recognition of the individual alphanumeric characters. Again, in Section 2 the Authors show that they focus only on recognizing the letters & numbers, but then -as I also stated above-  the "class" must also be properly defined and explained from the beginning, and then also define properly the "imbalance" and explain why is it important for the combinations of characters (as mentioned again at p.4 line 107)

 - In section 2.2 (or maybe 2.3, OR in section 3), the Authors should include a schematic of their CNN, explain its features & connections, provide the mathematical equations behind its operation, and compare it with the other realizations mentioned in literature and/or used in practice by PCB manufacturers.

- The ending of the sentence at lines 118-119 ("extracting the proportions so that they can occupy the proportions") is totally unclear, must be rephrased and explained properly. Similary, what is meant by "one grid of the original grid distribution occupies 10%"? Do the Authors mean that the number of 'pixels'/squares occupied by the character's drawing/shape represent 10% of the entire grid surface area?
     Likewise, I do not understand what is meant by "one grid occupies 1% of the images"? Do the Authors mean that one shape/character represents 1% of all the characters detected in the entire data set???

- I believe the Authors should explain in text the meaning of the "Mean" and "Standard  deviation" on the X & Y axes of Fig.s 3 and 9-11. Mean and standard deviation of what parameter? Is it the "Pixel Distribution"? If yes, how exactly is this "Pixel Distribution" defined mathematically, e.g. (most probably) with respect to the grid mentioned in Section 2.3, but how exactly? By the way, how big is the grid? If we consider it to have m x n pixels/squares, i.e. m rows & n columns, how much is m and how much is n?
 It is also totally unclear to me how the colorful images in Fig.s 9-11 can ultimately lead to an unequivocal unique quantitative value of a descriptor which can be used for decisions for correctly assessing the performance of the investigated method. Again, some mathematical equations should be used here to define this descriptor, which (probably) could be indicated at the same place where I mentioned about the detailed description of the mathematics for the operation of the Authors' CNN, in which the output would then have to be indicated by an equation, as probably that output is ultimately the quantitative decisional descriptor of interest. As is, I don't believe the reader can find a logical reasoning describing and leading to/deducing the final output/descriptor used for this assessment of the Deep Learning method(s) based on these figures, how did they result in the accuracy values provided subsequently?

MINOR
 - ANY Figure or Table should not be located in the middle of a page and interrupt the fluency of the text, but all Figures/Tables on a page should be placed either at its top and/or its bottom (or occuppy a whole page if large or made up of many elements/images).

- Eliminate the blank spaces with blank lines at the bottom of pages 4, 7, 9, 10, and 13 of the paper, which are most probably due to an inattentive insertion of subsequent Figures/Tables in the text of the original WORD manuscript. Cut-paste text into these blank lines and/or shift-reposition the location of the respective Fig.s/Tables.

 - I strongly suggest the Authors to double check that all data shown in all graphs & Fig.s are of good quality and clearly readable. For instance, the numbers along the X & Y axes for the graphs in Fig.3 are so small that they are impossible to see at all. I suggest that the uthors use high quality/resolution images and also that they enlarge signiicantly the size of their pics, so that the Figure occuppies fully the entire width of the page.

   Once these corrections/additions are done, I believe that the Authors can forward the revised paper directly to the Editor for faster publication.
With best wishes,
The reviewer

Reviewer 2 Report

My main concern about the paper is missing literature review of undersampling (e.g. SMOTE [1]). Besides the missing experiments with well-known techniques of dataset selection and an extensive comparison I have also doubts about whether the paper fits the journal. I suggest the authors to either rewrite the paper with an extensive study of undersampling and compare their method to the existing methods or focus (in an applied way) on how to show this study is significant in the field of PCB manufacturing. 

1. Chawla, Nitesh V., Kevin W. Bowyer, Lawrence O. Hall, and W. Philip Kegelmeyer. "SMOTE: synthetic minority over-sampling technique." Journal of artificial intelligence research16 (2002): 321-357.

Reviewer 3 Report

The article presents a method of efficiently recognizing single characters printed on PCBs through deep learning, which reduces a large amount of data to a basic data set.

In the methods based on deep learning, the learning and validation set is an important element. The effectiveness of forecasting and the thesis formulated largely depends on the selection of criteria and the amount of data. Therefore, the universality of the algorithm has certain limitations. I propose to describe in more detail on what basis and what is the reason for the selection of input data for the given research problem.